# *Mycobacterium ulcerans* culture results according to duration of prior antibiotic treatment: A cohort study

Brodie Tweedale[1,2], Fiona Collier[1], Nilakshi T. Waidyatillake[2,3], Eugene Athan[1,2], Daniel P. O'Brien[2,4] *

**1** Geelong Centre for Emerging Infectious Diseases (GCEID), Deakin University, Geelong, Australia, **2** Department of Infectious Disease, Barwon Health, Geelong, Australia, **3** Allergy and Lung Health Unit, Melbourne School of Population and Global Health, University of Melbourne, Melbourne, Australia, **4** Department of Medicine and Infectious Diseases, Royal Melbourne Hospital, University of Melbourne, Melbourne, Australia

* Danielo@barwonhealth.org.au

**Data Availability Statement:** As the data may contain potentially identifiable information, the authors are not authorized by the Barwon Health Human Research Ethics Committee to publicly

## Abstract

*Mycobacterium ulcerans* disease is a necrotising disease of the skin and subcutaneous tissue and is effectively treated with eight-weeks antibiotic therapy. Significant toxicities, however, are experienced under this prolonged regimen. Here, we investigated the length of antibiotic duration required to achieve negative cultures of *M. ulcerans* disease lesions and evaluated the influence of patient characteristics on this outcome. *M. ulcerans* cases from an observational cohort that underwent antibiotic treatment prior to surgery and had post-excision culture assessment at Barwon Health, Victoria, from May 25 1998 to June 30 2019, were included. Antibiotic duration before surgery was grouped as <2 weeks, ≥2-<4 weeks, ≥4-<6 weeks, ≥6-<8 weeks, ≥8-<10 weeks and ≥10–20 weeks. Cox regression analyses were performed to assess the association between variables and culture positive results. Ninety-two patients fitted the inclusion criteria. The median age was 60 years (IQR 28–74.5) and 51 (55.4%) were male. Rifampicin-based regimens were predominantly used in combination with clarithromycin (47.8%) and ciprofloxacin (46.7%), and the median duration of antibiotic treatment before surgery was 23 days (IQR, 8.0–45.5). There were no culture positive results after 19 days of antibiotic treatment and there was a significant association between antibiotic duration before surgery and a culture positive outcome (p<0.001). The World Health Organisation category of the lesion and the antibiotic regimen used had no association with the culture outcome. Antibiotics appear to be effective at achieving negative cultures of *M. ulcerans* disease lesions in less than the currently recommended eight-week duration.

## Introduction

*Mycobacterium ulcerans* disease, known as the Buruli ulcer (BU), is a necrotising infection of the skin and subcutaneous tissue and can cause large ulcerative lesions [1]. BU has been reported in 33 countries worldwide, primarily in tropical and subtropical countries including

share a de-identified data set. Data is available upon request from the Barwon Health Human Research Ethics Committee via email (regi@barwonhealth.org.au) for researchers who meet the criteria for access to confidential data.

**Funding:** The authors received no specific funding for this work.

**Competing interests:** The authors have declared that no competing interests exist.

Côte D'Ivoire, Ghana and Bénin where the largest burden of disease is observed [2, 3]. It also occurs in temperate regions such as south-eastern Australia where an increase in incidence is being observed with for example the number of cases managed at a tertiary referral centre in Victoria doubling between 2005–2010 and 2011–2017 [4]. It is the third most common mycobacterial disease amongst immunocompetent people and can result in significant morbidities and chronic deformities [3, 5]. The World Health Organisation (WHO) categorises lesions according to severity determined by size, number and involvement of critical sites [6]. It is important to understand optimum treatment therapies that minimise complications and maximise cure of the disease.

Eight-weeks of rifampicin-based combination antibiotic therapy is the currently recommended treatment for *M. ulcerans* disease [6, 7], and surgery may be used in addition to antibiotics [8]. Despite the effectiveness of these antibiotic regimens [9, 10], severe toxicities including gastrointestinal intolerance, hepatitis and rash are experienced in over 20% of patients in Australian populations and almost 15% of those require hospitalisation to manage adverse effects [11]. The risk of experiencing treatment side-effects exists throughout the whole treatment course [11], with the median time to develop antibiotic complications being 28 days (IQR 17–45 days). A shortened antibiotic duration therefore has the potential to reduce the incidence of antibiotic complications.

Initial evidence to support reduced treatment duration for *M. ulcerans* disease has come from studies in murine models. Ji, et al. reported the sterilisation of *M. ulcerans* infected mice footpads after only four weeks of antibiotic therapy [12]; and a more recent study by Chauffour, et al. has demonstrated comparable results [13]. Previous work by Cowan, et al. reported the effectiveness of a reduced antibiotic duration in humans after achieving cure in all patients receiving between four and six weeks antibiotic therapy combined with surgery [14]. Additionally, all of five excised lesions from which tissue was cultured for *M. ulcerans* following 28–35 days of antibiotics were negative, suggesting that only 4–6 weeks of antibiotics may be required to cure lesions [14], noting that the sensitivity of *M. ulcerans* culture is estimated at about 50% [15]. More recently, O'Brien et al. investigated the efficacy of six weeks of antibiotic therapy against the recommended eight weeks for the least severe, WHO category 1 lesions [16]. Cure was reported in 100% of 53 patients in the six-week treatment group, further supporting the potential for reduced antibiotic durations for the treatment of *M. ulcerans* disease [12–14].

To better understand the minimum duration of antibiotics required to achieve cure of *M. ulcerans* disease lesions, we investigated the length of antibiotic duration required to achieve culture negativity in a group of patients that underwent both surgery and post-excision culture assessment; with the consideration of the influence of patient characteristics on this outcome.

## Materials and methods

### Data collection

**Cohort identification.** Analysis was performed on a subset of prospectively collected data of all *M. ulcerans* cases treated from May 25 1998 to June 30 2019 by Barwon Health staff; a tertiary hospital located in Geelong, Australia. Cases encompassed patients from endemic regions of South-eastern Victoria, including the Bellarine Peninsula, Mornington Peninsula and surrounds. Data was prospectively collected using Epi-Info 6 (CDC, Atlanta) [17].

**Inclusion criteria.** For inclusion in this analysis, patients with diagnosed *M. ulcerans* must have (i) had surgery, (ii) received rifampicin based antibiotic treatment prior to surgery, and (iii) had Mycobacterium cultures performed on the excised lesions. The cohort included 20 cases published in an earlier study from our group [14]. The type of surgery performed included wide excision, conservative excision, curette and debridement as previously

described [8]. A *M. ulcerans* case was defined as the presence of a lesion clinically suggestive of *M. ulcerans*, plus any of:

i.  a culture of *M. ulcerans* from the lesion

ii.  a positive IS*2404* polymerase chain reaction from a swab or biopsy of the lesion

iii.  histopathology of an excised lesion showing a necrotic granulomatous ulcer with the presence of acid-fast bacilli consistent with acute *M. ulcerans* infection

Mycobacterial cultures from surgical specimens were performed at Barwon Health microbiological laboratories using Lowenstein-Jensen media and incubated at 30˚C for 12 weeks. Specimens were sent immediately from surgery to the laboratory which was on-site at Barwon Health. Here they were kept at room temperature (20˚C) and within 1–6 hours of arrival were finely diced with a sterile scalpel blade, without decontamination or dilution, prior to plating a tissue aliquot of approximately 2 mm in diameter.

### Definitions used during the analysis

Duration of symptoms was defined as the time between symptom onset and diagnosis; site of lesion was defined as upper limb, lower limb and trunk; antibiotic duration was grouped as <2 weeks, ≥2-<4 weeks, ≥4-<6 weeks, ≥6-<8 weeks, ≥8-<10 weeks and ≥10–20 weeks.

### Data analysis

Statistical analyses were performed using StataIC 16 (StataCorp, Texas, USA) [18]. Summary statistics were tabulated to describe the cohort characteristics. If antibiotics were changed due to adverse effects, only initial antibiotic combinations were examined when analysing the association with antibiotic regimens in the analysis. A Pearson's chi-squared test was used to assess the relationship between antibiotic duration and culture positive outcome, and a Kaplan-Meier curve was used to demonstrate the cumulative incidence of positive *M. ulcerans* cultures according to the days of antibiotic treatment. Rates of positive *M. ulcerans* cultures associated with identified variables were described in 100-person days of antibiotics. A Cox regression model was then used to assess the crude hazard ratios of variables with positive *M. ulcerans* cultures. Then a multivariate Cox regression analysis was performed including the variables sex and age a priori and the variable immune suppression as the only other variable to show evidence of an association with positive cultures in the crude analysis (assessed by p ≤ 0.20). The p-values for assessing the strength of the association of each variable with positive *M. ulcerans* cultures, controlled for all the other variables in the model, were determined by the likelihood ratio test.

### Ethics approval and consent to participate

This study was approved by the Barwon Health Human Research and Ethics Committee (HREC No. 04/60). All previously gathered human medical data were analysed in a de-identified fashion.

## Results

### Cohort characteristics

From May 25[th] 1998 to June 30[th] 2019, lesions excised from 92 patients following antibiotic therapy were cultured and included in the study. This represented 13.5% of 679 patients who were treated with antibiotics and 40.2% of 229 patients who had surgery in addition to antibiotics in the Barwon Health cohort during this time. Baseline cohort characteristics are

**Table 1. Baseline cohort characteristics.**

| Treatment Cohort (n = 92) | | |
|---|---|---|
| | n (%) | Mean ± SD (Range) and Median (IQR) |
| **Gender** | | |
| • Male | 51 (55.4) | |
| • Female | 41 (44.6) | |
| **Age (years)** | | Median = 60 (IQR 28–74.5) |
| • 0–19 | 19 (20.7) | |
| • 20–59 | 26 (28.3) | |
| • ≥ 60 | 47 (51.1) | |
| **WHO Category of Lesions** | | |
| • 1 | 54 (58.7) | |
| • 2 | 16 (17.4) | |
| • 3 | 22 (23.9) | |
| **Lesion Type** | | |
| • Nodule | 4 (4.4) | |
| • Oedematous | 24 (26.1) | |
| • Plaque | 4 (4.4) | |
| • Ulcer | 60 (65.2) | |
| **Antibiotic duration (days)** | | Median = 23 (IQR 8–45.5) |
| **Lesion Site** | | |
| • Upper limb | 29 (31.5%) | |
| • Lower limb | 61 (66.3%) | |
| • Head/Trunk | 2 (2.2%) | |
| **Antibiotic regimen** | | |
| • RCla* | 44 (47.8%) | |
| • RCp** | 43 (46.7%) | |
| • Other | 5 (5.4%) | |

*Rifampicin + Clarithromycin

**Rifampicin + Ciprofloxacin

presented in Table 1. The median age of patients included in this study was 60 years (IQR, 28–74.5), and 51 (55.4%) were male. Nine (9.8%) patients had diabetes mellitus and 12 (13%) were immune suppressed due to disease or medication. None were known to be HIV positive. BCG status was not collected noting that since the 1980s it has not been included in routine vaccine schedules in the local population. The majority of lesions were classified as WHO category 1 lesions (58.7%, n = 54), followed by WHO category 3 (23.9%, n = 22) and WHO category 2 lesions (17.4%, n = 16). There were no cases associated with osteomyelitis. Forty-seven days (IQR, 35–76) was the median time from onset of symptoms to commencement of treatment. The first-line antibiotic regimens were rifampicin and clarithromycin in 44 (47.8%) cases, rifampicin and ciprofloxacin in 43 (46.7%) cases, and other regimens in 5 (5.4%) cases. The median duration of antibiotic treatment before surgery was 23 days (IQR, 8–45.5). Thirty-three (35.9%) required a split-skin graft and 8 (8.7%) a vascularised flap following surgery. Thirty-six (39.1%) of cases experienced paradoxical reactions. All patients were cured following treatment with no disease relapses.

## Number of weeks of antibiotic treatment and *M. ulcerans* culture result

The proportion of patients with a positive *M. ulcerans* culture following less than two weeks of antibiotics was 51.6% (n = 31). After two to four weeks of treatment only 27.3% of patients had a positive culture, and there were no culture positive results evident after 2.7 weeks (19 days) (Figs 1 & 2). A chi-square test of goodness-of-fit confirmed a significant association between antibiotic duration before surgery in weeks and a culture positive outcome (p < 0.001) (Fig 1).

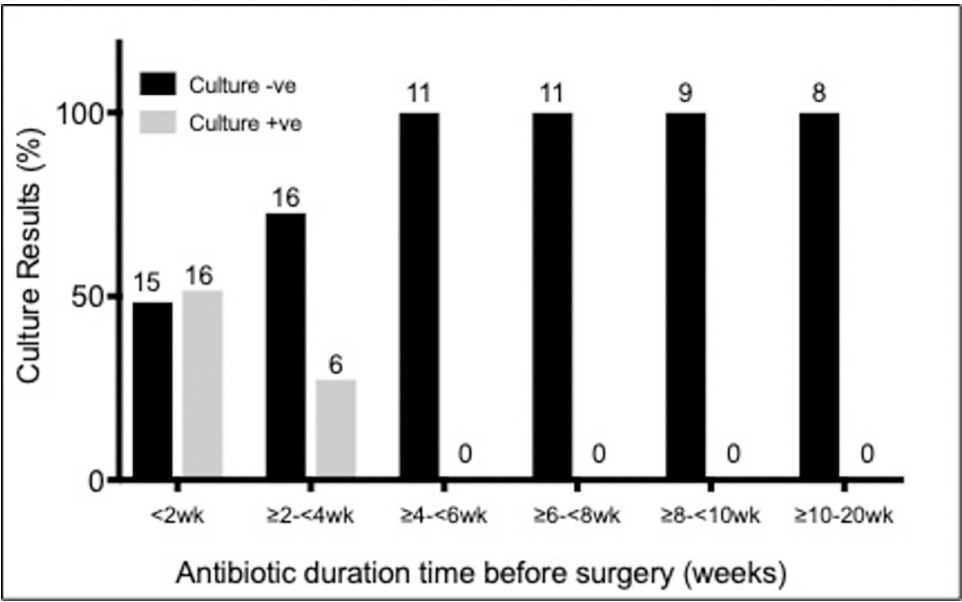

**Fig 1. Relationship between antibiotic duration before surgery (weeks) and culture results.** Figure labels on columns represent actual number of cases.

## Association between patient characteristics and *M. ulcerans* culture results

The association between patient characteristics and the outcome of the post-excision *M. ulcerans* culture was examined (Table 2). Using multivariable Cox regression model analysis adjusting for age, gender and immune suppression, there was a trend that older age (p = 0.07) and

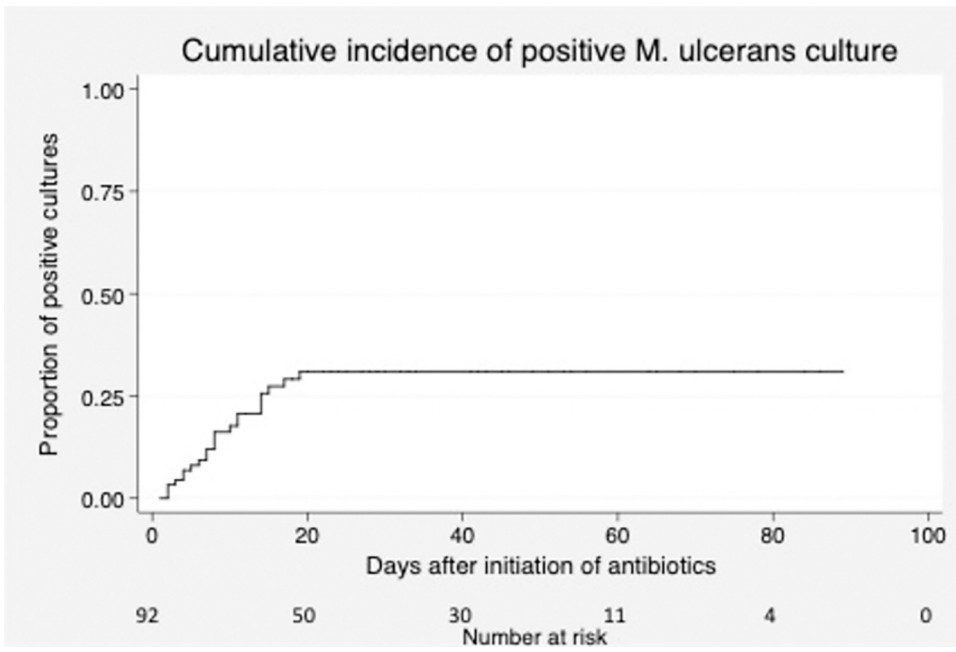

**Fig 2. Kaplan-Meier curve showing the cumulative incidence of positive *M. ulcerans* cultures according to days of antibiotic treatment.**

**Table 2. Cox regression model showing adjusted and unadjusted associations between identified variable and rates of positive *M. ulcerans* culture.**

| Variable | Failures (Positive culture) (%) | Follow-up (days) | Rate per 100-person days (95% CI) | Crude hazard ratio (95% CI) | p-value | Adjusted hazard ratio (95% CI) | p-value |
|---|---|---|---|---|---|---|---|
| **Gender** | | | | | | | |
| **Female** | 10 (24.4) | 1115 | 9.0 (4.83,16.67) | 1 | 0.97 | 1 | 0.86 |
| **Male** | 12 (23.5) | 1587 | 7.6 (4.5,14.1) | 1.0 (0.4,2.3) | | 0.9 (0.4,2.2) | |
| **Age (years)** | | | | | | | |
| **0–15** | 1 (7.1) | 603 | 1.7 (0.2–11.8) | 1 | 0.14 | 1 | 0.07 |
| **16–64** | 9(23.1) | 995 | 9.0 (4.7,17.4) | 4.1 (0.5,32.6) | | 4.3 (0.5,34.5) | |
| **≥65** | 12(30.8) | 1104 | 10.9 (6.2,19.1) | 5.1 (0.7,39.3) | | 6.6 (0.8,50.8) | |
| **WHO category of lesion** | | | | | | | |
| **1** | 14 (25.9) | 1160 | 12.1 (7.1,20.4) | 1 | 0.29 | - | - |
| **2** | 2 (12.5) | 703 | 2.8 (0.7,11.4) | 0.4 (0.1,1.6) | | - | |
| **3** | 6 (27.3) | 839 | 7.2 (3.2,15.9) | 0.9 (0.4,2.5) | | - | |
| **Lesion type** | | | | | | | |
| **Ulcer** | 15(25.0) | 1396 | 2.7 (2.1,3.4) | 0.7 (0.1–5.6) | 0.44 | - | - |
| **Nodule** | 1 (25.0) | 113 | 3.6 (1.6,8.0) | 1 | | - | - |
| **Oedema** | 6 (25.0) | 993 | 3.0 (1.6,5.8) | 0.7 (0.1,5.5) | | - | - |
| **Plaque** | 0 (0.0) | 200 | 0.0 | - | | - | - |
| **Lesion site** | | | | | | | |
| **Upper limb** | 9 (30.0) | 755 | 11.9 (6.2,22.9) | 1 | 0.38 | - | - |
| **Lower Limb** | 13 (21.3) | 1913 | 7.0 (3.9,11.7) | 0.6 (0.3,1.4) | | - | - |
| **Head/ Trunk** | 0 (0.0) | 34 | 0.0 | - | | - | - |
| **Immune suppression** | | | | | | | |
| **No** | 21 (26.3) | 2351 | 8.9 (5.8,13.7) | 1 | 0.16 | - | 0.06 |
| **Yes** | 1 (8.3) | 351 | 2.8 (0.4,20.2) | 0.3 (0.0,2.3) | | 0.2 (0,1.6) | |
| **Diabetes** | | | | | | | |
| **No** | 22 (27.2) | 2271 | 9.7 (6.4,14.7) | - | - | - | - |
| **Yes** | 0 (0.0) | 238 | 0 | - | | - | |
| **Antibiotic regimen** | | | | | | | |
| **RCla\*** | 9(20.5) | 1517 | 5.9 (3.1,11.4) | 1 | 0.48 | - | - |
| **RCp\*\*** | 12 (27.9) | 1026 | 11.7 (6.6,20.6) | 1.7 (0.7,4.0) | | - | |
| **other** | 1 (20.0) | 159 | 6.3 (0.9,44.6) | 1.0 (0.1,7.7) | | - | |
| **Weight (kg)** | | | | | | | |
| **0–90** | 4 (19.1) | 707 | 5.7 (2.1,15.1) | 1 | 0.71 | - | - |
| **≥ 90** | 2 (25.0) | 251 | 8.0 (2.0,31.9) | 1.3 (0.2,7.3) | | - | - |
| **Missing** | 16 (25.4) | 1744 | 9.2 (5.6,15.0) | 1.6 (0.5,4.7) | | - | - |
| **Duration of symptoms prior to diagnosis** | | | | | | | |
| **0–42** | 15 (27.8) | 1593 | 9.4 (5.7,15.6) | 1 | 0.50 | - | - |
| **≥42** | 6 (16.7) | 1030 | 5.8 (2.6,12.0) | 0.6 (0.2,1.6) | | - | - |
| **Missing** | 1 (50) | 79 | 12.7 (1.8, 89.9) | 1.7 (0.2,12.8) | | - | - |

\*Rifampicin + Clarithromycin

\*\*Rifampicin + Ciprofloxacin

lack of immune suppression (p = 0.06) were associated with a positive culture. However, there was no association between culture positivity and gender, WHO category, lesion type or site, diabetes, antibiotic regimen, weight or duration of symptoms prior to diagnosis.

## Discussion

Our results suggest BU lesions can be culture negative following antibiotic therapy significantly earlier than the recommended eight-weeks of treatment with no lesions culture positive after 19 days of treatment; a finding that supports the potential to reduce antibiotic treatment duration. Our study utilised data from a group of BU patients that had both rifampicin-based antibiotic treatment prior to surgical excision and *M. ulcerans* culture testing of the excised tissue. This enabled assessment of the antibiotic treatment duration needed for achieving culture negativity of lesions as well as assessing the influence of a number of patient covariates. We highlight the relationship between duration of antibiotic therapy and culture outcome, whereby an increased length of antibiotic therapy before surgery coincides with a decrease in the proportion of culture positive results.

Comparable results were yielded in an early study in Ghana in 2005, albeit in small numbers, where the cultures of 11 patients were negative after four-weeks of rifampicin-based therapy [19]. Additionally, an Australian study (n = 4) also demonstrated the inability to culture *M. ulcerans* from excised lesions following up to six weeks of antibiotic therapy [20]. Further, we showed in an earlier analysis of the Barwon Health cohort that 100% of lesions excised after 28–38 days of antibiotics were culture negative [14]. Importantly, four years on and with more than a four-fold increase in the number of patients in this specific cohort subpopulation (n = 92 compared to n = 20), the findings between these studies remain consistent, lending support to a reduced antibiotic duration for the treatment of *M. ulcerans* disease. Also, we recently reported that the cure of WHO category 1 lesions is achievable with just six-weeks of antibiotic therapy–affording further evidence for a reduced antibiotic duration [16]. Notably, the current study included nearly 40% of lesions which were WHO category 2 and 3, suggesting a shorter duration of antibiotic therapy may also be possible for more severe lesions.

Contrasting results have also been reported. In a study of patients treated with two weeks of rifampicin and streptomycin followed by six-weeks oral rifampicin and clarithromycin [21], three lesions (42.9%) were culture positive after 12-weeks from commencement of treatment. Additionally, in a randomised trial in Ghana, positive cultures were reported from three lesions (60%) after eight-weeks of antibiotic therapy [10]. Despite this, all lesions healed without any additional surgical intervention and no recurrence was reported after 12-month follow-up. This is suggestive that a reduced bacterial load, and not necessarily complete lesion sterilisation, may be adequate to permit an immune response strong enough to overcome mycolactone production and achieve wound healing. This was highlighted recently by O'Brien, et al. reporting a case series of five adults whose small ulcerative lesions spontaneously healed without specific treatment, emphasizing that a vigorous host immune response can overcome the suppressive effects of mycolactone and achieve cure [22].

A reduced antibiotic duration required to achieve cure will immediately benefit patients by reducing the daily inconvenience associated with antibiotic consumption. More importantly, the chance of experiencing drug-toxicities [11], the cost of treatment, and the impact on the microbiome will decrease [23]. This current study not only supports our previous proposition of a 25% reduction in the currently recommended eight-week treatment time for the sterilisation of selected small WHO category I BU lesions [16]; but also raises the possibility of an even further reduced antibiotic treatment time. This is significant as the median time for emergence of severe antibiotic complications is four weeks (IQR 17–45 days) [11]. Theoretically, over 50% of severe antibiotic complications could be prevented by reducing treatment time from eight-weeks to four-weeks. However, culture results of excised specimens don't necessarily correlate with clinical outcomes, especially as the sensitivity of *M. ulcerans* culture is estimated at about 50% [15]. Therefore, prospective randomised control trials comparing the reduced

duration against the recommended eight-week therapy should be conducted to assess the clinical effectiveness of shortened antibiotic regimens.

Other potential benefits of shortened antibiotic treatment regimens include a reduction in hospital and treatment fees associated with side-effects, as well as the decrease to the cost of antibiotics themselves; saving money for both patients and the health care system [24]. The potential effect on a patient's microbiome due to prolonged antibiotic use will also be reduced, therefore minimising disruptions to patient health [23, 25]. Additionally, the risk of antibiotic resistance will be reduced with a shortened treatment duration [26].

We found that there was some evidence that older age of the patient was associated with culture positivity (Table 2). We have recently reported that age greater than 65 years is a risk factor for experiencing more severe disease (Category 2 and 3); potentially due to reduced immunity and control of the bacteria [4, 27]. Thus it is also possible that a reduced immunity in older age slows the sterilising rate of lesions by antibiotics. Conversely, young individuals may be able to sterilise wounds more quickly through a greater penetration of wounds with antibiotics via improved tissue circulation. A possible association of increased culture positive results with no immune suppression is unexpected and the reasons for this are unclear.

We acknowledge that our study is limited by its observational nature, the fact that not all surgical specimens from our cohort were cultured and the relatively small sample size. Furthermore, postulations that *M. ulcerans* strains in Africa are more virulent due to production of increased quantities and more potent forms of mycolactone may limit the extrapolation of findings more globally [28]. However, we feel that the findings offer sufficient information to stimulate more research into the effectiveness of shortened antibiotic treatment regimens.

## Conclusions

We have shown in an Australian cohort that BU lesions can be culture negative following antibiotic therapy significantly earlier than the recommended eight-weeks of treatment. This provides the potential to significantly reduce toxicities, inconvenience and cost experienced by patients by reducing the duration of antibiotic treatment.

## Author Contributions

**Conceptualization:** Daniel P. O'Brien.

**Data curation:** Brodie Tweedale, Daniel P. O'Brien.

**Formal analysis:** Brodie Tweedale, Fiona Collier, Nilakshi T. Waidyatillake, Daniel P. O'Brien.

**Methodology:** Brodie Tweedale, Fiona Collier, Nilakshi T. Waidyatillake, Daniel P. O'Brien.

**Supervision:** Daniel P. O'Brien.

**Writing – original draft:** Brodie Tweedale.

**Writing – review & editing:** Fiona Collier, Nilakshi T. Waidyatillake, Eugene Athan, Daniel P. O'Brien.

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
