## [Decision Letter · Decision Letter 0]

19 Dec 2022

PONE-D-22-30052Mycobacterium ulcerans culture results according to duration of prior antibiotic treatment: a cohort studyPLOS ONE

Dear Dr. O'Brien,

Thank you for submitting your manuscript to PLOS ONE. After careful consideration, we feel that it has merit but does not fully meet PLOS ONE’s publication criteria as it currently stands. Therefore, we invite you to submit a revised version of the manuscript that addresses the points raised during the review process.

ACADEMIC EDITOR: This is an interesting 20-year cohort study on duration of antimicrobial therapy for patients with Buruli ulcer. While the premise is intriguing and the data seems robust, the reviewers have raised pertinent questions here that needs to be addressed. This article can be considered for publication after the aricle has been revised appropriately.  

We look forward to receiving your revised manuscript.

Kind regards,

Nitin Gupta, MBBS, MD, DM, AAHIVS, DTM&H

Academic Editor

PLOS ONE

Journal Requirements:

a) Did participants provide their written or verbal informed consent to participate in this study?

Additional Editor Comments:

This is an interesting 20-year cohort study on duration of antimicrobial therapy for patients with Buruli ulcer. While the premise is intriguing and the data seems robust, the reviewers have raised pertinent questions here that needs to be addressed. This article can be considered for publication after the aricle has been revised appropriately.

Reviewers' comments:

Reviewer's Responses to Questions

**Comments to the Author**

1. Is the manuscript technically sound, and do the data support the conclusions?

Reviewer #1: Yes

Reviewer #2: Yes

2. Has the statistical analysis been performed appropriately and rigorously? 

Reviewer #1: Yes

Reviewer #2: Yes

3. Have the authors made all data underlying the findings in their manuscript fully available?

Reviewer #1: Yes

Reviewer #2: Yes

4. Is the manuscript presented in an intelligible fashion and written in standard English?

Reviewer #1: Yes

Reviewer #2: Yes

5. Review Comments to the Author

Reviewer #1: Comments

Summary: the objective of the study is to investigate the length of antibiotic treatment duration required to achieve sterilization of M.ulcerans disease lesions. To reach such a target, the authors studied a cohort of 92 patients diagnosed with M.ulcerans disease who, from May 25, 1998 to June 30, 2019, had antibiotic treatment (mainly daily rifampicin + clarithromycin or ciprofloxacin) before surgery at Barton Health, Victoria, Australia and had culture of the excision lesions on Loewenstein Jensen medium.

Comment to the authors

The manuscript is well presented but, while the results of study are totally based on the sterilization or not of the excised lesions, almost no information is given on the technical means of the culture of excised specimens. It is only written “Mycobacterial cultures from surgical specimens were performed at Barwon Health microbiological laboratories using Lowenstein-Jensen media and incubated for 12 weeks (page 5, lines 129-130)”. More details are needed: time between lesion excision surgery and culture, temperature to which the excised lesions are kept, the interval of time between surgical excision and culture, mincing of the specimen, decontamination procedures if any, dilutions of the excision specimen, amount seeded on Loewenstein Jensen medium and culture temperature. All precisions on the culture procedures are decisive for the results interpretation. They are more important and far more decisive than all statistical and clinical details.

Page 6, lines 144-155 are Greek for the reviewer. Is it possible to make them simpler when the only important factor for interpretation of data and treatment recommendations is bacteriology and not statistics or clinics (5 pages on statistics and clinics versus two lines on bacteriology!)?

Page 11 line 210. The word “sterilized” is not appropriate. I suggest the change of wording. Negative culture of excision specimen is not at all synonymous of sterilization (as pointed out by the authors on page 13, line 260)

Reviewer #2: This paper describes the results of a 20-year prospective observational cohort relating to antibiotic duration prior to surgery for M. ulcerans diseases in Victoria, Australia, adding further evidence that shorter antimicrobial durations appear to be effective at sterilizing M. ulcerans disease lesions (under the 8 weeks recommendation). This paper is well-written and clear and contributes the literature on BU and its treatment. The paper, at times, lacks a few details, but I suspect that these may be corrected quite easily during revision. Specifically, the paper is limited by the absence of clinical outcomes, the lack of a more detailed description of the limited sensitivity of M. ulcerans culture and does not discuss the clinical and pathobiological diversity of BU/Mycolactone between Australia and West Africa. All told, I believe this paper should be published in PLOS ONE after the following revisions are addressed.

Suggested clarifications:

If possible, please describe when antimicrobial regimen changed, as only initial antimicrobial regimen was included.

Please include a mention of the sensitivity of culture in the introduction.

Please describe WHO classification of BU in the introduction as referred to multiple times in text. Consider mentioning that a minority of WHO I lesions may heal spontaneously.

Clarify number of WHO III associated with osteomyelitis.

Describe level of surgical debridement and if skin grafting occurred.

If possible, please comment on secondary infections.

I suspect that BCG is not administered in Victoria, Australia. As this is relevant to BU manifestation, please indicate whether this population received the BCG vaccine.

If possible, add whether paradoxical deterioration occurred during antimicrobial therapy.

The age discrepancy between BU in Australia (60s) and Africa (5-15 yo) is interesting. Please consider commenting on why BU is commonly seen in children in West Africa and older adults in Australia? Differences in water-related exposure/ behaviour? Mosquito-exposure in Australia?

Please comment on the differences in Mycolactone A and B (West Africa, possibly more virulent) and C (Australia, less virulent) in different geographic areas and possible differences in virulence, limiting global extrapolation of the findings.

Small comments:

Line 57: please add “accent circonflexe” to the o in Côte d’ivoire and accent aigu to Bénin.

Line 59: If possible, please provide numerical data regarding the increase in BU in South-eastern Australia.

Line 68: Please name the adverse drug effects. Hepatoxocity due to rifampin? Other drug effects?

Line 70: If possible, please provide duration of follow-up for this study. Eg: clinical cure and no relapse after X months of follow-up.

Line 85: 100% cure in how many patients? Please list sample size in this study.

Lines 173: please describe duration of follow-up for establishing that “no disease relapses” occurred.

Please describe most common other regimen 5%.

6. PLOS authors have the option to publish the peer review history of their article (what does this mean?). If published, this will include your full peer review and any attached files.

Reviewer #1: No

Reviewer #2: **Yes: **Carl Boodman

---

## [Author Response · Author response to Decision Letter 0]

2 Feb 2023

Dear Editor,

We are re-submitting our paper entitled “M. ulcerans culture results according to duration of prior antibiotic treatment: a cohort study” to be considered for publication as a research article in your journal. 

On the following pages please find in blue our responses to the reviewers’ comments. We have made the relevant changes in the revised copy of the manuscript which we hope addresses all of the concerns raised by the reviewers. We would be pleased to consider further revisions if needed.

All authors have seen and approved the manuscript and have contributed significantly to the work. The manuscript has not been published and is not being considered for publication elsewhere. Finally we have no conflicts of interest to disclose.

Yours sincerely

Associate Professor Daniel O’Brien MBBS, FRACP, DMedSc, Dip Anat.

 

Reviewer 1:

If possible, please describe when antimicrobial regimen changed, as only initial antimicrobial regimen was included. 

This has now been included in the methods section. Lines 161-162 

Please include a mention of the sensitivity of culture in the introduction.

This has now been included so that the section in the Introduction reads: “…suggesting that only 4-6 weeks of antibiotics may be required to cure lesions, noting that the sensitivity of M. ulcerans culture is estimated at about 50%.” Lines 91-92. 

Please describe WHO classification of BU in the introduction as referred to multiple times in text.

This information has now been included in the introduction as follows:

 “The World Health Organisation (WHO) categorises lesions according to severity determined by size, number and involvement of critical sites”. (Lines 66-67)

Consider mentioning that a minority of WHO I lesions may heal spontaneously.

This has been referred to in the discussion section (lines 277-280) as follows: “This was highlighted recently by O’Brien, et al. reporting a case series of five adults whose small ulcerative lesions spontaneously healed without specific treatment, emphasizing that a vigorous host immune response can overcome the suppressive effects of mycolactone and achieve cure[21]” 

Clarify number of WHO III with associated osteomyelitis.

This has been clarified in the results section as described: “There were no cases associated with osteomyelitis” Line 195.

Describe level of surgical debridement and if skin grafting occurred. 

A clarification of the types of surgery performed has now been included in the methods section as shown below, and the number and proportion of cases who underwent skin grafting or vascularised flap has now been included in the results section.

Methods, Lines 138-139: “The type of surgery performed included wide excision, conservative escision, curette and debridement….”

Results, Lines 199-200: “Thirty-three (35.9%) required a split-skin graft and 8 (8.7%) a vascularised flap following surgery.”

If possible, please comment on secondary infections? 

We did not collect this information so are unable to include it.

I suspect that BCG is not administered in Victoria, Australia. As this is relevant to BU manifestation, please indicate whether this population received the BCG vaccine.

This has now been included in the results section as follows:

Lines 192-193: “BCG status was not collected noting that since the 1980s it has not been included in routine vaccine schedules in the local population”

If possible, add whether paradoxical deterioration occurred during antimicrobial therapy. 

This information has now been included in the results section to read: Line 201: 

“Thirty-six (39.1%) of cases experienced paradoxical reactions.”

The age discrepancy between BU in Australia (60s) and Africa (5-15) is interesting. Please consider commenting on why BU is commonly seen in children in West Africa and older adults in Australia? Differences in water-related exposure/ behaviour? Mosquito-exposure? 

Whilst we agree this question is interesting, as it deals with epidemiology and risk of infection, rather than treatment, we feel it is out of scope for the current paper and have therefore not included a discussion on this issue. If the Editor still feels that it should be included, we can modify.

Please comment on the differences in Mycolactone A and B (West Africa) and C (Australia) in different geographic areas and possible differences in virulence, limiting global extrapolation of the findings.

This information has now been included in the limitations section of the manuscript as follows:

Lines 315-317: “Furthermore, postulations that M. ulcerans strains in Africa are more virulent due to production of increased quantities and more po¬tent forms of mycolactone may limit the extrapolation of findings more globally.”

Small comments:

Line 57: please add “accent circumflex” to the o in Côte d’ivoire and accent aigu to Bénin.

This has been corrected.

Line 59: If possible, please provide numerical data regarding the increase in BU in South_eastern Australia.

This has now been included in the introduction as follows: “…with for example the number of cases managed at a tertiary referral centre in Victoria doubling between 2005-2010 and 2011-2017.” Line 63-64

Line 68: Please name the adverse drug effects. Hepatoxocity due to rifampin? Other drug effects?

This has now been included as follows: “…severe toxicities including gastrointestinal intolerance, hepatitis and rash are experienced”. Line 74

Line 70: If possible, please provide duration of follow-up for this study. Eg: clinical cure and no relapse after X months of follow-up. 

For the study quoted, clinical outcomes were not described. It was a study looking at antibiotic complications and outcomes were censored at either the time of severe antibiotic complication or completion of the antibiotic course.

Line 85: 100% cure in how many patients? Please list sample size in this study.

This has now been included. Line 94

Reviewer 2:

The manuscript is well presented but, while the results of study are totally based on the sterilization or not of the excised lesions, almost no information is given on the technical means of the culture of excised specimens. It is only written “Mycobacterial cultures from surgical specimens were performed at Barwon Health microbiological laboratories using Lowenstein-Jensen media and incubated for 12 weeks (page 5, lines 129-130)”. More details are needed: time between lesion excision surgery and culture, temperature to which the excised lesions are kept, the interval of time between surgical excision and culture, mincing of the specimen, decontamination procedures if any, dilutions of the excision specimen, amount seeded on Loewenstein Jensen medium and culture temperature. All precisions on the culture procedures are decisive for the results interpretation. They are more important and far more decisive than all statistical and clinical details.

 We thank the reviewer for the suggestions. The requested detail has now been added to the methods section as follows: “Specimens were sent immediately from surgery to the laboratory which was on-site at Barwon Health. Here they were kept at room temperature (200C) and within 1-6 hours of arrival were finely diced with a sterile scalpel blade, without decontamination or dilution, prior to plating a tissue aliquot of approximately 2 mm in diameter.” Lines: 148-151

Page 6, lines 144-155 are Greek for the reviewer. Is it possible to make them simpler when the only important factor for interpretation of data and treatment recommendations is bacteriology and not statistics or clinics (5 pages on statistics and clinics versus two lines on bacteriology!)?

We have modified the section to reduce its length. However, as the majority of the section describes the statistical methods used to report many of the significant findings in our paper, we feel that it is important that the reader is able to determine exactly how these analyses were performed. Therefore we believe it is not helpful to the reader to simplify the section more than we have done. 

Page 11 line 210. The word “sterilized” is not appropriate. I suggest the change of wording. Negative culture of excision specimen is not at all synonymous of sterilization (as pointed out by the authors on page 13, line 260)

We have now modified the statement to say:

Lines 322-323: “….culture negative following antibiotic therapy”. We have also replaced the word ‘sterilized’ with ‘culture negative’ throughout the paper where appropriate.

---

## [Decision Letter · Decision Letter 1]

27 Mar 2023

Mycobacterium ulcerans culture results according to duration of prior antibiotic treatment: a cohort study

PONE-D-22-30052R1

Dear Dr. O'Brien,

We’re pleased to inform you that your manuscript has been judged scientifically suitable for publication and will be formally accepted for publication once it meets all outstanding technical requirements.

Kind regards,

Nitin Gupta, MBBS, MD, DM, AAHIVS, DTM&H

Academic Editor

PLOS ONE

Additional Editor Comments (optional):

The authors have addressed all the comments made by the reviewers. The article can be accepted in its current form.

Reviewers' comments:

Reviewer's Responses to Questions

**Comments to the Author**

1. If the authors have adequately addressed your comments raised in a previous round of review and you feel that this manuscript is now acceptable for publication, you may indicate that here to bypass the “Comments to the Author” section, enter your conflict of interest statement in the “Confidential to Editor” section, and submit your "Accept" recommendation.

Reviewer #1: All comments have been addressed

Reviewer #2: All comments have been addressed

2. Is the manuscript technically sound, and do the data support the conclusions?

Reviewer #1: Yes

Reviewer #2: Yes

3. Has the statistical analysis been performed appropriately and rigorously? 

Reviewer #1: Yes

Reviewer #2: Yes

4. Have the authors made all data underlying the findings in their manuscript fully available?

Reviewer #1: Yes

Reviewer #2: Yes

5. Is the manuscript presented in an intelligible fashion and written in standard English?

Reviewer #1: Yes

Reviewer #2: Yes

6. Review Comments to the Author

Reviewer #1: The manuscript and its conclusions are adequate for me . However I would suggest the authors (page 6, line143) "... to plating a tissue aliquot of approximatively 2mm in diameter" would deserve to be increased and tested.

Reviewer #2: All of my previous comments have been adequately addressed by the authors in this revised version. I believe this article is ready for publication.

7. PLOS authors have the option to publish the peer review history of their article (what does this mean?). If published, this will include your full peer review and any attached files.

Reviewer #1: **Yes: **Jacques Grosset

Reviewer #2: **Yes: **Carl Boodman

---

## [Editor Report · Acceptance letter]

14 Apr 2023

PONE-D-22-30052R1 

*Mycobacterium ulcerans* culture results according to duration of prior antibiotic treatment: a cohort study 

Dear Dr. O'Brien:

I'm pleased to inform you that your manuscript has been deemed suitable for publication in PLOS ONE. Congratulations! Your manuscript is now with our production department. 

Kind regards, 

on behalf of

Dr. Nitin Gupta 

Academic Editor

PLOS ONE